# Effects of *Coleus amboinicus* L. Essential Oil and Ethanolic Extracts on Planktonic Cells and Biofilm Formation of *Microsporum canis* Isolated from Feline Dermatophytosis

**DOI:** 10.3390/antibiotics11121734

**Published:** 2022-12-01

**Authors:** Arpron Leesombun, Karnchanarat Thanapakdeechaikul, Jiraporn Suwannawiang, Pipada Mukto, Sivapong Sungpradit, Norasuthi Bangphoomi, Tanasak Changbunjong, Orathai Thongjuy, Thekhawet Weluwanarak, Sookruetai Boonmasawai

**Affiliations:** 1Department of Pre-Clinic and Applied Animal Science, Faculty of Veterinary Science, Mahidol University, Nakhon Pathom 73170, Thailand; 2The Monitoring and Surveillance Center for Zoonotic Diseases in Wildlife and Exotic Animals (MoZWE), Faculty of Veterinary Science, Mahidol University, Nakhon Pathom 73170, Thailand; 3The Center of Veterinary Diagnosis, Faculty of Veterinary Science, Mahidol University, Nakhon Pathom 73170, Thailand

**Keywords:** *Microsporum canis*, *Coleus amboinicus*, essential oil, ethanolic extracts, antifungal, biofilm formation

## Abstract

*Microsporum canis* is an important zoonotic fungus that causes dermatophytosis in domestic animals and their owners. Domestic cats are the primary reservoir for *M. canis*. Antifungal drugs frequently produce adverse effects on the host animal, increasing the demand for novel alternative treatments derived from nature. We evaluated the antifungal activity of *Coleus amboinicus* essential oil (CEO) and ethanolic extracts (CEE) against *M. canis* in planktonic and biofilm growth. Twelve clinical isolates of *M. canis* were identified in feline dermatophyte samples. Using GC-MS, 18 compounds were identified in CEO, with carvacrol being the major constituent. HPLC analysis of CEE revealed that it contained rosmarinic acid, apigenin, and caffeic acid. The planktonic growth of all *M. canis* isolates was inhibited by *C. amboinicus* extracts. The minimum inhibitory concentration at which ≥50% of the isolates were inhibited (MIC_50_) was 128 µg/mL (32–256 µg/mL) for both CEO and CEE. The MIC_90_ values of CEO and CEE were 128 and 256 µg/mL, respectively. CEO at MIC (128 µg/mL) and 2× MIC (256 µg/mL) significantly inhibited the biofilm formation of weak, moderate, and strong biofilm-producing *M. canis*. CEE at 2× MIC (256 µg/mL) significantly inhibited the biofilm formation of all isolates. Overall, *C. amboinicus* extracts inhibited planktonic growth and exhibited a significant antibiofilm effect against *M. canis*. Thus, *C. amboinicus* is a potential source of natural antifungal compounds.

## 1. Introduction

*Microsporum canis* is an important zoophilic dermatophyte in domestic cats and dogs; however, it also causes dermatophytosis in humans. Human infections are acquired from domestic animals through direct contact with clinically or subclinically infected animals [1]. *M. canis* is a major species of keratinophilic and keratinolytic filamentous fungi that cause superficial fungal infections worldwide [2], particularly in Europe, the eastern Mediterranean, and South America [3,4,5]. *M. canis* infections in cats, dogs, and other domestic animals such as rabbits generally manifest as multifocal alopecia, scaling, and circular lesions [6,7]. Both stray and domesticated cats are considered common reservoirs for *M. canis* [8,9,10] and anyone in direct contact with these animals, including owners and veterinarians, is at risk of becoming infected [9]. Treatments include oral and topical preparations of antifungal agents such as amphotericin B, griseofulvin, terbinafine, itraconazole, flucytosine, and fluconazole, administered alone or in combination. These agents cause adverse reactions, such as nephrotoxicity, hepatotoxicity, and neurotoxicity and have been associated with treatment failure secondary to drug resistance [11,12]. One antifungal-resistant strain of *M. canis* is resistant to terbinafine and has been reported in some feline patients [13]. Factors mitigating drug resistance include a lack of patient compliance, poor tissue penetration, and variable medication bioavailability [11,14]. Biofilms produced by *M. canis* are an additional concern [15,16]. Biofilm refers to the matrix that surrounds the microbial population and acts as a physical barrier, protecting the microorganisms in it. Biofilms are typically associated with increased resistance to antifungal agents [15].

Plants, including their extracts, contain phytochemical compounds that are used as antimicrobials and antibiofilm agents [17]. Identifying novel antifungal targets and compounds derived from natural plants will help develop novel antifungal strategies and improve existing ones [15]. Several previous (and ongoing) studies have focused on the use of plant extracts as alternative treatments for fungal infections [12]. Plant extracts that have antifungal activity include lemon grass (*Cymbopogon citrates*), lantana (*Lantana camara*), nerium (*Nerium oleander*), basil (*Ocimum basilicum*), and olive leaves (*Olea europaea*) [18]. The ethanolic extracts of *Echinophora platyloba* inhibit *Candida albicans* ATCC 10231 [19]. The aqueous and ethanolic extracts and the essential oil of *Thymus capitatus* exhibit antifungal activity against *C. albicans* and *M. canis* [20]. Extracts of *Ocimum gratissimum* leaves display antifungal activity against *M. canis, Microsporum gypseum, Trichophyton rubrum,* and *Trichophyton mentagrophytes* [21]. Methanolic leaf extracts of *Eucalyptus camaldulensis* were investigated for suspected *in vitro* antifungal activities against *M. canis*, *M. gypseum*, *T. rubrum*, *Trichophyton schoenleinii*, *T. mentagrophytes*, and *Epedermophyton floccosum* [22].

*Coleus amboinicus* Lour. (synonym: *Plectranthus amboinicus* [Lour.] Spreng), whose common name is Indian borage, is widely cultivated in tropical Africa, Asia, and Australia. This perennial herb belonging to the family Lamiaceae contains several phytochemicals, including monoterpenoids, diterpenoids, triterpenoids, sesquiterpenoids, phenolics, flavonoids, and esters [23,24]. The geographic location of *C. amboinicus* cultivation influences the phytochemical composition of plant extracts [25]. The stem, leaf, and root extracts of *C. amboinicus* contain high concentrations of polyphenolics such as caffeic acid, rosmarinic acid, apigenin, chrysoeriol, 5-*O*-methyl-luteolin, and 5,8-dihydroxy-7,2′,3′,5′-tetramethoxyflavone, which is a novel flavonoid [26,27,28]. Several studies have reported the wide range of pharmacological properties of *C. amboinicus* extracts and essential oil, including antioxidant activity [25], antidandruff action [29], antiproliferative effects on cancer cells [30], analgesic and anti-inflammatory activities [31], antirheumatoid arthritis [32], anti-inflammatory effect following bone injury [33], antibacterial activity against methicillin-resistant *Staphylococcus aureus* [34], mosquitocidal and water sedimentation properties [35], and insect repellent [36] and insecticidal effects [37].

Although *C. amboinicus* exhibits antifungal activity against several fungal species [38], little is known of its antifungal effects on dermatophytes and the biofilms they produce. Therefore, we evaluated the antifungal effects of *C. amboinicus* essential oil (CEO) and *C. amboinicus* ethanolic extracts (CEE) against planktonic cells and biofilm formation of *M. canis* clinically isolated from a feline dermatophyte.

## 2. Results

### 2.1. Chemical Composition of CEO and CEE

The yield of essential oil obtained from fresh *C. amboinicus* leaves was 0.08% (*v*/*w*), and the oil was clear and light yellow. The refractive index, density (g/cm^3^), and specific gravity of the oil at 20 °C were 1.505, 0.935, and 0.937, respectively. Figure 1 presents the chromatogram of the main components of CEO. Gas chromatography–mass spectrometry (GC-MS) analysis revealed 18 compounds, representing 99.84% of the total composition of CEO (Table 1). The mass spectrum of each compound is presented in Appendix A. The concentrations of carvacrol, β-caryophyllene, and thymol in CEO were determined and found to be 3.4 ± 0.2, 0.35 ± 0.16, and 0.013 ± 0.08 mg/mL, respectively.

The yield of CEE was 2.2% *w*/*w*, appearing as a dark green and highly viscous solid. The total phenolic content was 666.0 ± 9.1 mg gallic acid equivalent (GAE)/g sample and the total flavonoid content was 462.3 ± 3.1 mg quercetin equivalent (QE)/g sample. The three compound standards, namely rosmarinic acid, apigenin, and caffeic acid, were qualified via high-performance liquid chromatography ([HPLC]; Figure 2). The concentrations of rosmarinic acid, apigenin, and caffeic acid used for quantification were 1.251, 1.175, and 0.732 mg/g samples, respectively (Table 2).

### 2.2. Fungal Isolation and Biofilm Formation

We identified 12 *M. canis* isolates based on morphology, polymerase chain reaction (PCR), and gene sequencing. A total of 720 base pair (bp) amplicons were obtained from positive *M. canis* samples using targeted ITS1-5.8S-ITS2 PCR. Bidirectional DNA sequencing was performed to demonstrate that all tested feline specimens were *M. canis* ITS1-5.8S-ITS2 sequences, which shared 100% sequence identity with those isolated from Thai and Belgium cats (sampled from hair and skin) and are deposited in relevant databases (MT487850 and OW988573). Additionally, the sequences identified in this study were perfectly matched with *M. canis* ITS1-5.8S-ITS2 sequences isolated from dog (ON527777 and KT155637), horse (LC623726 and OW984765), rabbit (OW987260 and OW987262), and human (OW988577) specimens (Appendix A).

All 12 *M. canis* isolates were classified as weak (25%), moderate (50%), or strong biofilm producers (25%) based on their biofilm production. *C. albicans* and *T. rubrum* were classified as strong and moderate biofilm producers, respectively (Table 3). The biofilm production of *M. canis* is shown in Appendix A.

### 2.3. Effect of CEO and CEE on Planktonic Cells

CEO and CEE inhibited the planktonic cell growth of all *M. canis* isolates. The results of the minimum inhibitory concentration (MIC) analysis of CEO and CEE are presented in Table 3 and Table 4. The minimal fungicidal concentration (MFC) and MIC values of each treatment were comparable. CEO exhibited antifungal activity comparable to that of CEE, with an MIC_50_ of 128 µg/mL (32–256 µg/mL). The MIC_90_ of CEE was twice that of CEO. Fluconazole had an MIC_50_ and MIC_90_ of 8 µg/mL (4–16 µg/mL) and 16 µg/mL, respectively (Table 4).

### 2.4. Effect of CEO and CEE on Biofilm Formation

CEO at MIC (128 µg/mL) and 2× MIC (256 µg/mL) significantly inhibited the biofilm formation of *C. albicans*, *T. rubrum*, and all weak, moderate, and strong biofilm-producing *M. canis* isolates (Figure 3). CEE at MIC (128 µg/mL) significantly inhibited the biofilm formation of *C. albicans*, *T. rubrum*, and all weak (100%), five moderate (83.3%), and two strong (66.7%) biofilm-producing *M. canis* isolates. CEE at 2× MIC (256 µg/mL) significantly inhibited the biofilm formation of all *M. canis* isolates (Figure 4). Fluconazole at MIC (8 µg/mL) and 2× MIC (16 µg/mL) did not affect the biofilm formation of *C. albicans* and *T. rubrum*.

## 3. Discussion

To the best of our knowledge, this is the first report of the antifungal activity of *C. amboinicus* extracts against planktonic cells and the biofilm formation of *M. canis*. *M. canis* biofilms are composed of a multidirectionally expanded network of hyphae linked together by a polysaccharide extracellular matrix [16]. Biofilm reduces the penetrability of antifungal agents, thus contributing to treatment failure and recurrent infection [39,40]. The inhibitory effect of antifungal agents on biofilm formation was observed at concentrations higher than those required to inhibit the growth of planktonic cells [41]. Fungal biofilm formation is a key factor in fungal virulence, persistence, and invasion as well as recurrent fungal infections and conventional antifungal resistance [40,42]. The time-dependent adherence of arthroconidia was observed, starting at 2 h and up to 6 h after inoculation. *M. canis* produced keratinolytic enzymes and secreted endo and exoproteases during adhesion; this process was likely inhibited by chymostatin, a serine protease inhibitor [43]. After biofilm formation for 72 h, a polysaccharide extracellular matrix that links fungal hyphae was observed [16,44]. The extracellular matrices of poor, moderate, and strong biofilm-producing *M. canis* appear to be related to mechanisms of antifungal resistance; however, further investigations are needed to confirm this. Flucytosine or fluconazole treatment at every 6–24 h could not completely destroy the biofilms of *Candida* spp. Poor drug penetration might not be a major mechanism of antifungal resistance for *Candida* biofilms [45]. During the early stage of *C. albicans* biofilm formation, genes encoding efflux pumps are upregulated, thereby mediating antifungal resistance [46]. Developing new compounds or alternative inhibitors to treat biofilm-related drug resistant fungal infections is essential to veterinary and human medicine [40,42,47]. In this study, fluconazole (4–16 µg/mL) had no effect on the mature biofilms of *M. canis* isolates. This result was similar to those reported by Bila et al., who found that fluconazole only inhibited the metabolic activity of early-stage biofilms of *T. mentagrophytes* at 32 mg/L but did not exhibit antibiofilm activity on mature biofilms, even at the highest concentration (512 mg/mL) [48].

Our study also demonstrated that *C. amboinicus* can inhibit planktonic cell growth and biofilm formation of feline zoonotic *M. canis*. CEO and CEE significantly inhibited the planktonic cell growth of *M. canis* at 128 µg/mL (32–256 µg/mL). Considering the MIC_90_ values, CEO was found to have a higher potency more potent than CEE against all *M. canis* isolates (MIC_90_ of 128 µg/mL vs. 256 µg/mL). *C. amboinicus* has been reported to exhibit antifungal activity against several fungi, including *Aspergillus clavatus*, *Aspergillus niger*, *Cladosporium cladosporioides*, *Chaetomium globosum*, *Myrothecium verrucaria*, *Penicillium citrinum*, *Trichoderma viride*, and *Mucor* sp. [38,49]. It also inhibits the biofilm formation of other pathogenic microorganisms, such as *Streptococcus mutans, Streptococcus pyogenes,* and *S. aureus* [50,51,52]. *C. amboinicus* is rich in monoterpenes, including carvacrol, thymol, eugenol, chavicol, and ethyl salicylate [37,49,53]. The 18 compounds identified in CEO in this study represent 99.84% of the total essential oil and include carvacrol (56.65%), p-cymene (10.89%), and γ-terpinene (9.33%); these three compounds alone comprise 76.87 % of the total essential oil. The concentrations of carvacrol, β-caryophyllene, and thymol were found to be 3.4 ± 0.2, 0.35 ± 0.16, and 0.013 ± 0.08 mg/mL, respectively. This result differs from that reported by da Costa et al., who found thymol to be the major constituent (64.3%), followed by p-cymene (10.3%), γ-terpinene (9.9%), and β-caryophyllene (2.8%) [54]. Previous studies have reported that the phytochemical composition of CEO is significantly influenced by the cultivation location, processes, and methods of essential oil extraction [25,37]; for example, steam distillation produced higher levels of carvacrol in *C. amboinicus* essential oil than those produced via the hydrodistillation method [37]. CEO at MIC had excellent effects against all clinical isolates. The high potency of CEO may be attributed to the hydrophobic property of essential oil, which adversely affects every step of biofilm formation, including adhesion, growth, maturation, and dissemination. The antibiofilm mechanisms of essential oil include reducing bacterial adhesions, preventing fresh biofilm formation, and destroying existing biofilm [55,56].

Carvacrol appears to be a major contributor to the antifungal properties of CEO. Carvacrol, p-cymene, and γ-terpinene are monoterpenes that exhibit various biological activities, including antioxidative, anti-inflammatory, anxiolytic, antineoplastic, and antimicrobial effects [57]. The antimicrobial effects of carvacrol are effective against various microorganisms, including bacteria such as *S. aureus* and *Pseudomonas aeruginosa* and fungi such as *C. albicans*, *Candida glabrata*, and *Candida parapsilosis* [58]. In a previous study, *Lavandula multifida* L. essential oil containing carvacrol as the main constituent was effective against dermatophytes (MIC: 0.16 μL/mL) and *Cryptococcus neoformans* (MIC: 0.32 μL/mL) [59]. Carvacrol also exhibits antifungal activity against *Aspergillus* spp. (MIC: 100 µg/mL) and *Cladosporium* spp. (MIC: 100 µg/mL) [60,61], possibly by binding to sterols in the fungi. The sterols residing on planktonic cell membranes are essential for their survival [62], and their hydrophilic properties allow carvacrol to penetrate the polysaccharide layer of their biofilm matrix, thereby destabilizing the biofilm [63]. A recent study reported that p-cymene exhibited no antifungal activity against *A. niger* (MIC: >300 µL/mL) and *Rhizopus oryzae* (MIC: >1024 µg/mL) [64], whereas γ-terpinene has been shown to exhibit antifungal activity against *Sporothrix schenckii* (MIC = 62.5–500 µg/mL) and *Sporothrix brasiliensis* (125–500 µg/mL) [65].

In the present study, CEE effectively eradicated the biofilm formation of weak, moderate, and strong biofilm producers at 2× MIC. Total phenolic and flavonoid contents were positively correlated with the antimicrobial activity of the plant extracts [66]. We found higher total phenolic and flavonoid levels in CEE than those reported in previous studies. For example, the ethanolic extract of *C. amboinicus* leaves obtained from Vietnam had a total phenolic and total flavonoid content of 26.84 ± 0.91 µg GAE/mg sample and 12.14 ± 0.42 µg QE/mg sample, respectively [67]. A methanolic extract of the *C. amboinicus* stem obtained from India had a total phenolic content of 49.91 mg GAE/g sample and total flavonoid content of 26.6 mg rutin equivalent/g sample [68]. Flavonoids inhibit nucleic acid biosynthesis and spore germination in plant pathogens [69,70]. High phenolic and flavonoid levels may thus be related to the significant antifungal effects of CEE. Importantly, CEE contained remarkable levels of rosmarinic acid (1.251 mg/g sample), apigenin (1.175 mg/g sample), and caffeic acid (0.732 mg/g sample) in this study. Rosmarinic and caffeic acid compounds have significant antifungal effects against *Fusarium oxysporum* [71]. The antifungal mechanism of rosmarinic acid is poorly understood but is believed to be related to the RTPase enzyme [72]. Apigenin at a concentration of 5 µg/mL exhibited antifungal activity against *C. albicans*, *C. parapsilosis*, *Malassezia furfur*, *T. rubrum*, and *T. beigelii* by inhibiting biofilm formation and efflux-mediated pumps of fungi. It also induced cell death by interfering with membrane function and increasing cell permeability [73,74]. Mice infected with *T. mentagrophytes* recovered after treatment with apigenin ointment administered at concentrations of 2.5 and 5 mg/g on the 12th and 16th days, respectively [75]. Caffeic acid phenethyl ester, a major active component of propolis (*Apis trigona*), has been shown to exert concentration-dependent effects on planktonic cells and biofilm formation of different *Candida* species [76] and synergistically enhance the antifungal activity of fluconazole against resistant clinical isolates of *C. albicans* [77]. Another study reported that the fungicidal activity of caffeic acid against *T. rubrum* was observed at 86.59 μM; this activity was mediated via plasma membrane damage and reduced ergosterol production, where caffeic acid reduced isocitrate lyase activity and downregulated critical genes (*ERG1*, *ERG6*, and *ERG11*) required for ergosterol synthesis [78].

Although CEO and CEE had different chemical constituents, both exhibited excellent and comparable inhibitory activities against all fungal isolates obtained from feline dermatophyte samples. Our findings suggest that both CEO and CEE act as natural antifungal agents against planktonic cells and biofilm-producing *M. canis*. Future investigations of the relationship between plant-based compounds, such as carvacrol in CEO and apigenin in CEE, their mechanisms of action, and classification based on biofilm production may contribute to a better understanding and guide the development of safe and effective antifungal agents derived from natural sources.

## 4. Materials and Methods

### 4.1. Plant Preparation and Extractions

*C. amboinicus* Lour. was harvested from a pesticide-free garden in the Nonthaburi Province, Thailand (13.862162, 100.409385). The plants were identified and housed at the herbarium within the Faculty of Pharmacy, Mahidol University, Thailand. The voucher specimen was PBM-005507-08. The hydrodistillation method was used to process the fresh leaves of *C. amboinicus* for essential oil extraction. The extraction was performed using a Clevenger-type apparatus operating at atmospheric pressure. The collected CEO was dried with anhydrous sodium sulfate, transferred to amber glass bottles, and stored at 40 °C. The physical properties of the CEO, including color, density, refractive index, and specific gravity, were evaluated and recorded. The yield of CEO was determined based on the weight of the fresh plant material before processing and was expressed in % (*v*/*w*) [37].

During the ethanolic extraction process, *C. amboinicus* leaves were dried in a hot air oven at 60 °C for 72 h and ground into small pieces. The leaf fragments were macerated in 95% ethanol at room temperature (RT) for 5 days. The extract solution was filtered through sterile gauze and a vaporized solvent using a rotary evaporator at 40 °C (BÜCHI, Flawil, Switzerland). The CEE was lyophilized in Labconco FreeZone 4.5 L Freeze Dryer equipped with Lyo-Works™ Operating System (Labconco, Kansas City, MO, USA) and stored at −20 °C. The yield of CEE was determined based on dry weight, weight after lyophilization, and weight of the leaf fragments before processing, and expressed in % (*w*/*w*).

### 4.2. GC-MS

The chemical composition of CEO was analyzed via GC-MS. The samples were diluted with methanol and injected in the split mode (1:10 split ratio) into the GC-MS model Agilent 7890A/5977B GC/MSD system equipped with a DB-5HT capillary column (0.1 µm film thickness × 0.25 mm diameter × 30 m length; Agilent Tech., Santa Clara, CA, USA) at a flow rate of 1 mL·min^−1^ in helium (carrier gas) and an injector temperature of 250 °C. The initial oven temperature was 40 °C (5 min), which was then increased to 250 °C at a rate of 10 °C/min and maintained there for 5 min. The following MS settings were used: ion source temperature, 230 °C; ionization energy, 70 eV; and mass scan range, 35–550 *m*/*z*. Compounds were identified by matching their mass spectra against those specified in Wiley Registry 7th Edition MS libraries. The concentrations of the major components were calculated by comparing the peak area of samples with the peak area of standard compounds.

### 4.3. Determination of Total Phenolic Content

The total phenolic content of CEE was determined using Folin–Ciocalteu’s colorimetric assay, with slight modifications. The stock extract solution (1000 μg/mL) was mixed with 125 µL of Folin–Ciocalteu reagent (Merck, Darmstadt, Germany) in a 1:1 ratio for 5 min. Subsequently, 400 µL of 7.5% sodium carbonate was added to the mixture, followed by incubation at RT for 30 min. The absorbance of the final mixture was measured at 760 nm using the Synergy H1 Hybrid Multi-Mode Microplate Reader (BIOTEK, Winooski, VT, USA). Gallic acid was used to prepare the standard curve (with a 40–240 μg/mL calibration range). The gallic acid solutions and the results are expressed in GAE/g of the crude extracts [79].

### 4.4. Determination of Flavonoid Content

The modified aluminum chloride colorimetric method was used to determine the flavonoid content of the plant extracts [80]. A 250-µL aliquot of the extract solution (1000 μg/mL) was mixed with 1.25 mL of deionized water, after which a 5% sodium nitrite solution (75 µL) (Sigma Aldrich, St. Louis, MO, USA) was added and the mixture was allowed to stand for 5 min. Subsequently, 150 µL of 10% aluminum chloride (Sigma Aldrich, St. Louis, MO, USA) was added to the extract solution, followed by 500 µL of 1 M sodium hydroxide. The solution was further diluted with 275 µL of deionized water and allowed to stand for 6 min. Finally, the absorbance was measured at a wavelength of 510 nm using the Synergy H1 Hybrid Multi-Mode Microplate Reader (BIOTEK, Winooski, VT, USA). A quercetin solution (30–300 μg/mL) was used to prepare the standard calibration curves.

### 4.5. HPLC

CEE was analyzed using HPLC. The HPLC 1290 Infinity II system: Zorbrax Eclipse Plus C18 column (2.1 × 50 mm, 1.8-Micron; Becton, Dickinson and Company, Franklin Lakes, NJ, USA) with an ultraviolet (UV) detector (280 nm) was used at a gradient flow of 0.5 mL/min. The mobile phase composition was 3% acetic acid in water: 1% acetic acid in water acetonitrile. The injection volume was 2 μL. The column temperature was maintained at 30 °C. The stock extract solution (10 mg/mL) was dissolved in methanol and filtered through 0.45-µm nylon membrane filters before performing HPLC. The compounds present in the extracts were characterized according to their UV–vis spectra and identified in terms of their retention time relative to that of known standards: rosmarinic acid, apigenin, and caffeic acid (Sigma Aldrich, St. Louis, MO, USA) [25]. A standard graph was generated using standard solutions of 5–500 µg/mL. The regression equation correlating to the area under the peak (Y) and standard (X) was as follows: Y = 5.04 (X: rosmarinic acid) + 4.37, Y = 8.98(X: apigenin) + 0.31 and Y = 13.04 (X: caffeic acid) + 1.31.

### 4.6. Sample Collection and Fungal Identification

The study protocol was approved by the Faculty of Veterinary Science, Animal Care and Use Committee (MUVS-2019-09-45) and the Faculty of Veterinary Science-Institutional Biosafety Committee (IBC/MUVS-B-005-2562). Skin, nail, and hair specimens were randomly collected from cat patients with feline dermatophytosis during 2019–2020 at Prasu-Arthorn Animal Hospital, Faculty of Veterinary Science, Mahidol University, Thailand. The samples were placed on Difco™ Potato Dextrose Agar (PDA) (Becton, Dickinson and Company, NJ, USA) plates supplemented with 0.1% chloramphenicol. The *M. canis* isolates were screened based on the morphology of the colonies, including their size, texture, and color. The characteristics, size, and arrangement of microconidia and macroconidia were evaluated by lactophenol cotton blue staining and observed under a light microscope at 10× and 40× magnification [81].

### 4.7. Confirmation of M. canis Using Molecular Techniques

PCR was used to confirm the species of *M. canis*. Twelve fungal samples obtained from feline patients were grown on PDA at 27 °C for 7 days before DNA extraction using the QIAamp DNA Micro Kit (Qiagen, Hilden, Germany). Samples stored at −80 °C were used to determine DNA concentration and purity using the Beckman Coulter DU 730 Uv/Vis Spectrophotometer (Beckman Coulter, Brea, CA, USA).

PCR was performed using the universal fungal primers of the ITS1-5.8S-ITS2 gene, i.e., ITS1: 5′-TCCGTAGGTGAACCTGCGG-3′ and ITS4: 5′-TCCTCCGCTTATTGATATG-3′ [82,83]. The reaction mixture (25 µL) contained 12.5 µL of GoTaq^®^ Green Master Mix (Promega, Madison, Wisconsin, USA), 10 pmol of ITS1 and ITS4 primers, and 100 ng of DNA template and ultrapure water (Milli-Q; pH 6.5). PCR was performed under the following conditions: 94 °C for 2 min; 35 cycles of 94 °C for 30 s, 55 °C for 60 s, and 72 °C for 90 s; the final extension was performed at 72 °C for 10 min. Subsequently, 3 µL of PCR products was mixed with 3 µL of loading dye and 3 µL of GelStar™ Nucleic Acid Gel Stain (Lonza, Basel, Switzerland) and examined via 1.5% agarose gel electrophoresis and visualized via UV irradiation. Six (50% of each biofilm classification) of the amplified PCR products (720 bp) were cleaned using the ExoSAP-IT™ PCR Product Cleanup Reagent (Thermo Fisher, Waltham, MA, USA) according to the manufacturer’s protocol. The PCR products were amplified by mixing them with a 5× sequencing buffer (Thermo Fisher, Waltham, MA, USA). The amplification was performed at 96 °C for 1 min, 96 °C for 10 s for 25 cycles, 50 °C for 5 s, 60 °C for 4 s, and finally cooling at 15 °C. The nucleotide sequences were determined using the NextSeq 550 sequencing system (Illumina Inc., San Diego, CA, USA) and compared to reference strains from the GenBank^®^ database (National Center for Biotechnology Information, Bethesda, MD, USA).

### 4.8. Bioinformatic and Phylogenetic Analysis

All *M. canis* ITS1-5.8S-ITS2 sequences were compared with sequences available in the GenBank database using The Basic Local Alignment Search Tool (http://blast.ncbi.nlm.nih.gov/Blast.cgi: accessed on 18 August 2022). Multiple alignments of all nucleotide sequences were conducted using the ClustalW web-based tool (https://www.genome.jp/tools-bin/clustalw: accessed on 18 August 2022) [84]. The final dataset comprised 387 positions. Phylogenetic trees were reconstructed using maximum likelihood analysis with bootstrapping (100 replications) in the advanced mode of the phylogeny.fr web server (http://www.phylogene.fr/: accessed on 18 August 2022) [85]. Published sequences in the GenBank database originating from other global locations were used to compare all sequences. *S. schenckii* (MG976612), *T. rubrum* (MK027017), *C. albicans* (OW988269), and *C. neoformans* var. *neoformans* (KP068909) were included as the outgroup for the ITS1-5.8S-ITS2 phylogenetic tree. The *M. canis* ITS1-5.8S-ITS2 nucleotide sequence data obtained in this study are available in GenBank using the accession numbers OP227140–OP227145.

### 4.9. Biofilm Formation Classification

All feline dermatophyte samples were grown in Sabouraud dextrose agar (SDA) plates (Becton, Dickinson and Company, Franklin Lakes, NJ, USA) until conidia formation was identified. The fungal inoculum was prepared by covering the colonies with 5 mL of 0.85% sterile saline and gently scraping the colony’s surfaces with a sterile cotton swab. The conidial suspension was transferred into sterile tubes, which were allowed to stand for 5–10 min to allow for sedimentation of the hyphae. The supernatant, which contained microconidia and macroconidia, was transferred into new sterile tubes. Turbidity was adjusted to 0.5 McFarland (approximately 2 × 10^6^ CFU/mL), and the microconidia and macroconidia were counted using a hemocytometer [86].

The biofilm formation assay was performed as previously described [41]. Briefly, 200 μL of each fungal inocula was added to sterile 96-well polystyrene plates (Kartell S.p.A., Noviglio, MI, Italy) and incubated at 37 °C for 3 h to allow for adhesion. The wells were then washed with 200 μL of phosphate-buffered saline (PBS) at a pH of 7.0; then, 200 µL of RPMI 1640 (Sigma Aldrich, St. Louis, MO, USA) was added. The plates were incubated at 37 °C for 72 h without shaking. Each fungal isolate preparation was performed in triplicate. *C. albicans* ATCC 90028 and *T. rubrum* (reference strain from the Center of Veterinary Diagnosis, Faculty of Veterinary Science, Mahidol University, Thailand) were used as the experimental controls.

Each biofilm biomass was quantified using crystal violet assay. The planktonic cells were discarded, and the attached cells were gently washed twice with PBS. After drying the plates at RT for 10 min, the cells were fixed with 200 μL of absolute methanol for 10 min and subsequently dried for 10 min. After 10 min of drying, 100 μL of aqueous 0.3% crystal violet solution was added to each well and the plates were incubated at RT for 20 min. The remaining dye was removed, and the biofilms were washed with PBS to remove any excess dye. After drying for another 10 min, the crystal violet that had accumulated in the biofilm cells was decolorized using 150 μL of 33% acetic acid for 30 s. Finally, each solution was transferred to a new plate, and the optical density was measured immediately at 590 nm using the BIOTEK ELx808 spectrophotometer (BIOTEK, Winooski, VT, USA) [87].

The biofilm formation cutoff was established according to optical density “ODc,” defined as the mean absorbance at 590 nm of control RPMI 1640. “OD” refers to the optical density of the treated biofilm solution. Biofilm formation ability was classified as none (OD ≤ ODc), weak (ODc < OD ≤ 2× ODc), moderate (2× ODc < OD ≤ 4× ODc), or strong (4× ODc < OD) [87].

### 4.10. Effects of CEO and CEE on Planktonic Cells

The microdilution method (based on standard Clinical and Laboratory Standards Institute Guidelines, 2008) [88], was used to determine the MIC. CEO and CEE were mixed with 100 mg/mL dimethyl sulfoxide in a RPMI 1640 medium supplemented with L-glutamine, which was buffered to pH 7.0 with 3-[N-morpholino] propane sulfonic acid (Sigma Aldrich, MO, USA) [41,48], and then subjected to two-fold serial dilution to obtain concentrations from 1024 µg/mL to 2 µg/mL. Fluconazole (Sigma Aldrich, St. Louis, MO, USA), an antifungal agent used as the positive control, was prepared at the same concentration as CEO and CEE. Subsequently, 100 µL of each concentration was added into the wells, followed by inoculation with 100 µL of each fungal suspension to obtain a final concentration of 2 × 10^3^ CFU/mL. All procedures were performed in triplicate. Culture media was used as the negative control. *C. albicans* and *T. rubrum* were used for internal quality control. The microplates were incubated at 35 °C for 96 h. The MIC was defined as the lowest concentration of extract at which no visible growth was observed under an inverted microscope.

MFCs were established by streaking the subcultures taken from the MIC wells without visible growth on an SDA plate. After incubation at 37 °C for 96 h in aerobic conditions, viable fungal growth was evaluated. The MFC was defined as the lowest concentration of extract at which no fungal growth was observed under an inverted microscope.

### 4.11. Effects of CEO and CEE on the Biofilm Formation of M. canis

The susceptibility of mature *M. canis* biofilms to CEO and CEE was evaluated by exposing the biofilms to extracts at MIC and 2× MIC. The extracts were diluted in RPMI 1640 and added to each well of a 96-well plate containing a mature biofilm, following which the plates were incubated at 37 °C for 96 h. Each treatment was performed in triplicate. After incubation, the biofilm metabolic activity of the fungi was determined via XTT (2,3-bis(2-methoxy-4-nitro-5-sulfophenyl)-2H-tetrazolium-5-carboxanilide) reduction assay [89]. XTT (Sigma Aldrich, St. Louis, MO, USA) was prepared according to the manufacturer’s instructions, and 100 μL of XTT-menadione solution was added to each well. The plates were then incubated in the dark for 1–2 h at 37 °C, after which absorbance at 490 nm was measured using the Synergy H1 Hybrid Multi-Mode Microplate Reader (BIOTEK, Winooski, VT, USA). The results are expressed as the average of absorbance values.

### 4.12. Statistical Analysis

Data were evaluated for normal distribution using the Shapiro–Wilk test prior to one-way analysis of variance. All statistical analyses were performed using IBM SPSS Statistics version 21.0. A *p* value of < 0.05 was considered statistically significant.

## 5. Conclusions

The increasing resistance in zoonotic fungi and adverse reactions to antifungal agents are major challenges for the development of natural-based agents. We found CEO and CEE to be extremely effective against the planktonic cell growth of clinical *M. canis* isolates. *C. amboinicus* also had significant antibiofilm effects on weak, moderate, and strong fungal biofilm producers. Thus, *C. amboinicus* may emerge as a novel source of natural antifungals. The antifungal mechanisms of *C. amboinicus* and drug formulations warrant further investigation for developing safe and effective treatments for zoonotic *M. canis* infections.

## Figures and Tables

**Figure 1 antibiotics-11-01734-f001:**
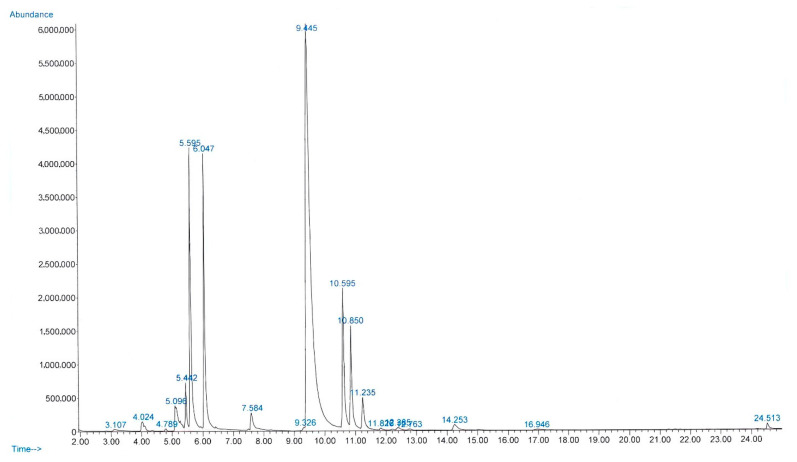
Chromatogram of the main components of *Coleus amboinicus* essential oil, as determined via gas chromatography–mass spectrometry.

**Figure 2 antibiotics-11-01734-f002:**
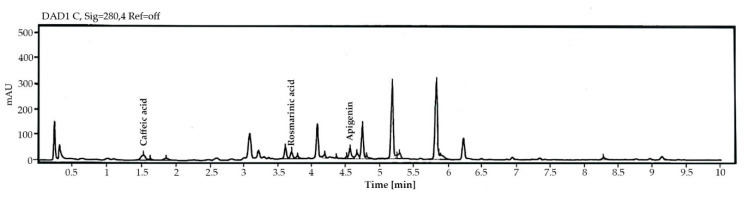
High-performance liquid chromatography of *Coleus amboinicus* ethanolic extracts.

**Figure 3 antibiotics-11-01734-f003:**
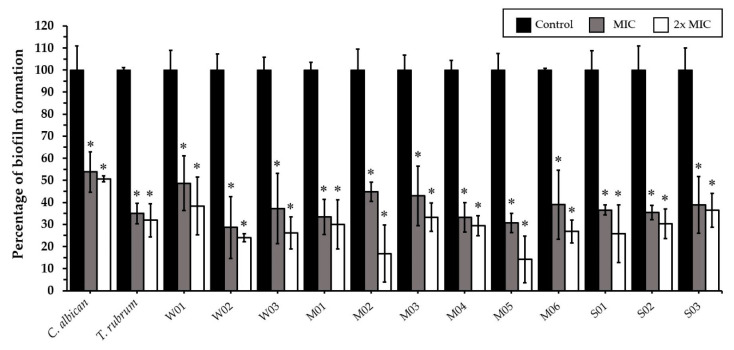
Effects of *Coleus amboinicus* essential oil on the biofilm formation of *Candida albicans* ATCC 90028, *Trichophyton rubrum,* and *Microsporum canis* after 96 h of treatment. Each bar represents the mean ± SD of three experiments per group. * differences between the control and treatments were statistically significant (*p* < 0.05).

**Figure 4 antibiotics-11-01734-f004:**
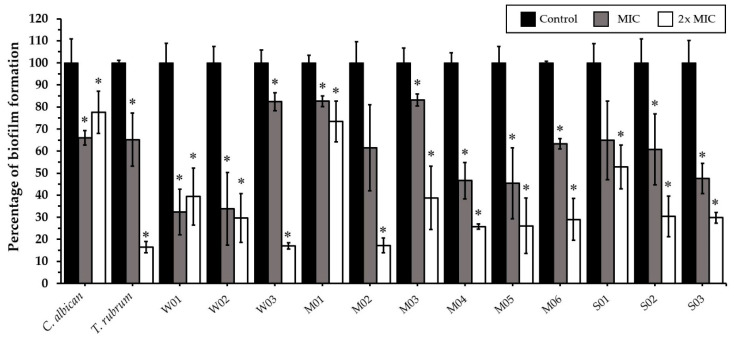
Effects of *Coleus amboinicus* ethanolic extracts on the biofilm formation of *Candida albicans* ATCC 90028, *Trichophyton rubrum,* and *Microsporum canis* after 96 h of treatment. Each bar represents the mean ± SD of three experiments per group. * differences between the control and treatments were statistically significant (*p* < 0.05).

**Table 1 antibiotics-11-01734-t001:** Chemical composition of *Coleus amboinicus* essential oil.

No.	Retention Time (min)	Classes	Compounds	Formula	Chemical Structure	Peak Area (%)	Similarity Index (%)
1	3.10	Aldehydes	Hexenal	C_6_H_10_O	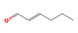	0.17	94
2	4.02	Monoterpene	α-Thujene	C_10_H_16_	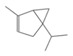	0.71	94
3	4.78	Monoterpene	Sabinene	C_10_H_16_	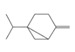	0.12	91
4	5.09	Monoterpene	β-Myrcene	C_10_H_16_	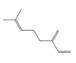	2.57	97
5	5.44	Monoterpene	α-Terpinene	C_10_H_16_	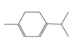	1.62	98
6	5.59	Monoterpene	p-Cymene	C_10_H_14_	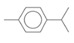	10.89	97
7	6.04	Monoterpene	γ-Terpinene	C_10_H_16_	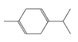	9.33	97
8	7.58	Monoterpene	4-Terpineol	C_10_H_18_O	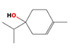	1.26	96
9	9.32	Monoterpene	Thymol	C_10_H_14_O	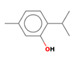	0.17	90
10	9.44	Monoterpene	Carvacrol	C_10_H_14_O	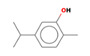	56.65	97
11	10.59	Sesquiterpene	β-Caryophyllene	C_15_H_24_	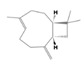	7.12	99
12	10.85	Sesquiterpene	α-Farnesene	C_15_H_24_	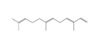	5.41	91
13	11.23	Sesquiterpene	α-Humulene	C_15_H_24_	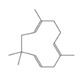	2.28	97
14	11.82	Sesquiterpene	β-Farnesene	C_15_H_24_	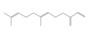	0.19	92
15	12.38	Sesquiterpene	β-Bisabolene	C_15_H_24_	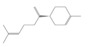	0.26	95
16	12.76	Sesquiterpene	β-Sesquiphellandrene	C_15_H_24_	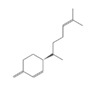	0.11	95
17	14.25	Sesquiterpene	(-)-Caryophyllene oxide	C_15_H_24_O	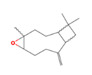	0.69	74
18	24.51	Phthalates	Bis(2-ethylhexyl) phthalate	C_24_H_38_O_4_	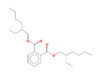	0.29	91
Total						99.84	

**Table 2 antibiotics-11-01734-t002:** Retention time, peak area, and concentration of compounds identified in *Coleus amboinicus* ethanolic extracts using high-performance liquid chromatography.

No.	Retention Time (min)	Compounds	Peak Area	Concentration (mg/g Sample)
1	1.52	Caffeic acid	96.78	0.73
2	3.70	Rosmarinic acid	67.43	1.25
3	4.55	Apigenin	105.93	1.17

**Table 3 antibiotics-11-01734-t003:** Biofilm classification and minimum inhibitory concentration (MIC) of CEO, CEE, and fluconazole against planktonic cells of fungal isolates.

No.	Fungi	Biofilm Classification	MIC (µg/mL)		
			Fluconazole	CEO	CEE
1	*Candida albicans*	strong	8	128	64
2	*Trichophyton rubrum*	moderate	4	32	256
3	W01	weak	8	32	128
4	W02	weak	8	128	128
5	W03	weak	16	128	128
6	M01	moderate	4	32	128
7	M02	moderate	4	256	128
8	M03	moderate	8	64	128
9	M04	moderate	8	128	128
10	M05	moderate	8	128	256
11	M06	moderate	16	128	32
12	S01	strong	4	64	256
13	S02	strong	4	64	256
14	S03	strong	8	128	64

CEO, *Coleus amboinicus* essential oil; CEE, *C. amboinicus* ethanolic extracts.

**Table 4 antibiotics-11-01734-t004:** MIC_50_, MIC_90_, GM of MIC, and MIC range of fluconazole, *Coleus amboinicus* essential oil (CEO), and *C. amboinicus* ethanolic extracts (CEE) against planktonic cells of *Microsporum canis*.

Treatment	Minimum Inhibitory Concentration (MIC) (µg/mL)			
	MIC_50_	MIC_90_	GM of MIC	MIC Range
Fluconazole	8	16	8	4–16
CEO	128	128	90.51	32–256
CEE	128	256	128	32–256

MIC_50_, the minimum inhibitory concentration at which ≥50% of the isolates were inhibited; MIC_90_, the minimum inhibitory concentration at which ≥90% of the isolates were inhibited; GM: geometric mean.

## Data Availability

The data presented in this study are available within the article.

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
