# Peer review of "Effects of Coleus amboinicus L. Essential Oil and Ethanolic Extracts on Planktonic Cells and Biofilm Formation of Microsporum canis Isolated from Feline Dermatophytosis"

_antibiotics, 2022, doi:10.3390/antibiotics11121734_

Round 1

Reviewer 1 Report

The study “The effects of Coleus amboinicus L. essential oil and ethanolic 2 extracts on planktonic cells and biofilm formation of Microsporum canis isolated from feline dermatophytosis” form Leesombun et al. explores an interesting topic. But in order to increase the usefulness and significance of the study, it needs a revision before being considered suitable for readers and there are some points to overcome for acceptance.

In this article, author found that C. amboinicus extracts inhibited planktonic growth and exhibited a significant antibiofilm effect against M. canis. However, the introduction section is quite plain, not in deep. It is recommended to tone up the introduction section.

Picture quality is poor in figure 1 and 2.

Line 92 C. amboinicus should be in italics.

Conclusion is very simple and small recommended to tone up.

It is suggested a moderate English revision by an English native speaker in order to polish text from typos and imperfections.

Unwanted spacing and typo mistakes throughout the manuscript. Need to be check and correct carefully.

Double check the way of adding references in the main text body and reference section as per journal guidelines.

Reviewer 2 Report

I appreciate the authors efforts in doing this study. The result of this study are potentially be good but the representation of the study does not reflect that at all. Moreover, it requires thorough revision and additional data before it can be considered for publication. Please see below some of my major concerns:

For me any biofilm study must be supported by microscopy images (SEM, Confocal or Fluorescence etc.). Please provide atleast 1 image to see the inhibition. Crystal violet is a gold standard but an initial step method. Or atleast show a microtitre plate with biofilms with inhibition and add in supplementary file.

Detail about Coleus amboinicus is very less in the introduction. The whole work is on Coleus amboinicus, but its significance, importance and ethnopharmacological information is missing.

Table 1: Please add extra two columns for adding class of the compounds and structure

Please provide mass spectra for each compound in supplementary file.

Table 3: On what basis biofilm classification is decided? How did you calculated the strength of biofilm production? Nothing is mentioned about that.

In section 4.1, why the leaves were dried in oven and that also for 72 hours. This is too much. It should have been air dried for better results. Oven drying for this length of time will effect the phytochemical content and incur an increase in loss rate of phytocompounds. This is not correct.

Figure 3 and 4 mentioned absorbance on the axis. It is better to portray your results in %inhibition. This will be easy to understand that how much % of biofilm is inhibited after treatment. Absorbance will prove nothing.

It is difficult to understand that how conidia was attached to the surface and how biofilm was produced. It needs to be shown by microscopic images.

How did you made sure that there was no bacterial contamination in the plates, as you have not used any antibacterial drug. So, there might be a possibility that the biofilm produced in the plates are because of some bacteria and not fungi. 

There are no negative control in the work. So it is difficult to rely on such results.

Authors can elaborate the discussion and introduction on plant-based natural antibiofilm compounds for better significance and impact by citing these articles.

https://doi.org/10.3390/plants11050610

https://doi.org/10.3390/life12101616

Reviewer 3 Report

The manuscript ‘The effects of Coleus amboinicus L. essential oil and ethanolic extracts on planktonic cells and biofilm formation of Microsporum canis isolated from feline dermatophytosis’ is a comprehensive report that discusses the inhibitory effect of Coleus amboinicus on the growth of the fungi, Microsporum canis.

Major comments

Suggest the future directions of the research at the end of the discussion. For instance, suggest some of the individual components from the extracts that could be used to inhibit the biofilm formation of Microsporum canis based on the understanding of the authors.

Consider discussing the effect of apigenin and caffeic acid as well

Biofilms have different stages such as adhesion, maturation, and dissemination. And the biofilm is stronger at the later stages. But the treatment results are only provided for a specific time period. A time course result would have been more appropriate. Substantiate why mature biofilms are used in this study.

MIC values were lower for strong biofilms and higher for weak biofilms. Discuss the discrepancies in the manuscript.

Why is the disk diffusion susceptibility test not considered in the present study?

Why are compounds of high abundance such as p-Cymene and γ-Terpinene in Coleus amboinicus essential oil not discussed in the manuscript?

Figure 3 results are not discussed well enough

Include the reference for the hydrodistillation method

Why is the inhibitory effect results of the extracts on planktonic cells not included in the manuscript

Minor comments

Include the time period of the treatment in the figure caption

Lines 19 – 21 could be moved to the introduction section from abstract

Line 86 consider rephrasing the sentence

C. amboinicus is not italicized in some places. Also, check the supplementary material

Line 129 Weak biofilm producers (25%) and strong biofilm producers (25%). There's a typo

Line 429 protocol says 490 nm

Round 2

Reviewer 1 Report

Author addressed all comments carefully so I endorse this article for the publication

Author Response

Reviewer 1

Point 1:  Author addressed all comments carefully so I endorse this article for the publication. English language and style are fine/minor spell check required.

Response 1: Thank you for your advice. We are grateful for your insightful comments on our paper. English language and style have been totally revised.

Reviewer 2 Report

Manuscript is significantly improved by the authors. However, some minor revision is still required before it can be considered for publication. 

In Table 1, class of the compounds is still missing. Please add additional column and show the class of each compounds. Furthermore, suggested work in the previous revision can be cited.

Author Response

Reviewer 2

Manuscript is significantly improved by the authors. However, some minor revision is still required before it can be considered for publication.

Point 1: In Table 1, class of the compounds is still missing. Please add additional column and show the class of each compounds. Furthermore, suggested work in the previous revision can be cited.

Response 1: We have added the classification of compounds in table 1 at Line 108. And the suggested article has been cited at line 60.

Reviewer 3 Report

The manuscript is thoroughly revised and the authors have addressed all the comments meticulously. However, the manuscript requires thorough proofreading by a native speaker.

Author Response

Reviewer 3

Point 1: The manuscript is thoroughly revised and the authors have addressed all the comments meticulously. However, the manuscript requires thorough proofreading by a native speaker.

Response 1: Thank you for this suggestion. Therefore, I have sent the revised manuscript to be totally re-edited by Enago’s Copyediting service. The main changes are marked up using the Track Changes function in the manuscript.
